# Microglia are required for developmental specification of AgRP innervation in the hypothalamus of offspring exposed to maternal high-fat diet during lactation

Haley N Mendoza-Romero, Jessica E Biddinger, Michelle N Bedenbaugh, Richard Simerly*

Department of Molecular Physiology & Biophysics, Vanderbilt University, Nashville, United States

## eLife Assessment

The authors provide a **valuable** contribution by documenting the role of microglia in pruning the axon terminals of AgRP neurons. The analysis of microglial axonal pruning is **solid**; however, the analysis of the effects inhibiting microglia on subsequent food consumption is not fully complete.

**\*For correspondence:**
richard.simerly@vanderbilt.edu

**Competing interest:** The authors declare that no competing interests exist.

**Abstract** Agouti-related peptide (AgRP) neurons in the arcuate nucleus of the hypothalamus respond to multiple metabolic signals and distribute neuroendocrine information to other brain regions such as the paraventricular hypothalamic nucleus (PVH), which plays a central role in metabolic homeostasis. Neural projections from AgRP neurons to the PVH form during the postnatal lactational period in mice and these projections are reduced in offspring of dams that consumed a high-fat diet (HFD) during lactation (MHFD-L). Here, we used immunohistochemistry to visualize microglial morphology in MHFD-L offspring and identified changes that were regionally localized to the PVH and appeared temporally restricted to the period when AgRP neurons innervate this region. In addition, axon labeling experiments revealed that microglia engulf AgRP terminals in the PVH, and that the density of AgRP innervation to the PVH in MHFD-L offspring may be dependent on microglia, because microglial depletion blocked the decrease in PVH AgRP innervation observed in MHFD-L offspring, as well as prevented the increased body weight exhibited at weaning. Together, these findings suggest that microglia are activated by exposure to MHFD-L and interact directly with AgRP axons during postnatal development to permanently alter innervation of the PVH, with implications for developmental programming of metabolic phenotype.

## Introduction

Maternal nutritional status has a profound effect on the metabolic phenotype of offspring. Children born to obese mothers experience higher rates of obesity later in life, with accompanying comorbidities that negatively impact health and longevity (*Stettler et al., 2005*; *Whitlock et al., 2009*; *Tamashiro and Moran, 2010*; *Andersen et al., 2012*). Although this developmental programming of metabolic phenotype has been reproduced in several animal models (*Samuelsson et al., 2008*; *Masuyama and Hiramatsu, 2014*; *García-Cáceres et al., 2019*; *Skowronski et al., 2024*), the underlying mechanisms remain poorly defined. In mouse models, maternal obesity during lactation, a time when offspring are dependent on milk from their mothers for nutrition, appears to be particularly impactful. These changes to metabolic phenotype are thought to be mediated by changes in the milk

(*Gorski et al., 2006*; *Vogt et al., 2014*; *Calvo-Lerma et al., 2022*) and occur without subsequent dietary challenge to the offspring themselves, suggesting that they are a consequence of developmental programming (*Bolton et al., 2022*; *Skowronski et al., 2024*). Because neural circuits known to control body weight develop during the lactational period, they are vulnerable to a variety of environmental signals that may affect their organization and function (*Horvath et al., 2010*; *Bouret et al., 2015*; *Elson and Simerly, 2015*; *Zeltser, 2018*; *Skowronski et al., 2024*).

AgRP neurons in the arcuate nucleus of the hypothalamus (ARH) function as 'hunger neurons' that respond to key metabolic signals such as leptin, ghrelin, glucose and free fatty acids (*Krashes et al., 2011*; *Betley et al., 2015*; *Chen et al., 2015*; *Sutton Hickey et al., 2023*), and they distribute this information to other regions associated with energy balance regulation (*Simerly, 2008*; *Zagmutt et al., 2018*). Thus, the ability of AgRP neurons to influence other components of feeding circuitry is dependent on the formation of their neural connections, which develop primarily during the first 2 wk of postnatal life (*Bouret et al., 2004a*). During development, AgRP axons extend from the ARH at postnatal day 4 (P4) and reach the PVH between P8-P10. Leptin is required for normal targeting of AgRP axons to downstream regions, and in leptin-deficient mice both neuroanatomical and related physiological defects persist into adulthood (*Bouret et al., 2004b*; *Bouyer and Simerly, 2013*). Maternal overnutrition affects formation of feeding circuits during postnatal life with concomitant dysregulation of body weight (*Plagemann et al., 1992*; *Lippert and Brüning, 2022*; *Skowronski et al., 2024*). Limiting HFD exposure of dams to the first 3 weeks of lactation (MHFD-L) causes suppression of neural projections from AgRP neurons to the PVH in offspring and is associated with increased body weight later in life (*Vogt et al., 2014*). In fact, MHFD-L was more effective than prenatal maternal HFD exposure in causing body weight changes in adult offspring. MHFD-L did not alter cell number, peptidergic expression, or cellular activity of AgRP neurons in the ARH, suggesting that maternal nutritional status during lactation is particularly important for the establishment of neural connections related to the control of body weight. Notably, the effects of both leptin (*Kamitakahara et al., 2018*) and MHFD-L (*Vogt et al., 2014*) on targeting AgRP projections display considerable regional specificity.

Adult mice placed on HFD display a marked hypothalamic gliosis that reveals an acute inflammatory response, which presages significant weight gain (*Horvath et al., 2010*; *Valdearcos et al., 2017*; *Spencer et al., 2019*; *Cansell et al., 2021*). This hypothalamic neuroinflammation is characterized by marked changes in the density and morphology of microglia that are most pronounced in the ARH (*Thaler et al., 2012*; *Valdearcos et al., 2014*). Microglia are the resident myeloid cells of the CNS and respond to a broad array of circulating signals, including nutrients such as saturated fats and carbohydrates (*Valdearcos et al., 2014*; *Nadjar et al., 2017*; *Leyrolle et al., 2019*; *Butler et al., 2020*). Moreover, activation of microglia alone is sufficient to stimulate food intake and promote weight gain in adult mice, and perturbations that block activation of microglia reduce the metabolic disruption associated with neuroinflammation (*Valdearcos et al., 2017*; *Rosin and Kurrasch, 2019*; *Sun et al., 2024*). Because of their established role as nutrient-sensing sentinels of hypothalamic neuroinflammation, and their documented participation in neural development (*Stevens et al., 2007*; *Schafer et al., 2012*; *Stephan et al., 2012*), microglia have been proposed as possible mediators of developmental programming caused by nutritional perturbations (*Rosin and Kurrasch, 2019*; *Folick et al., 2021*).

Evidence from several lines of investigation indicates that microglia have multiple roles in brain development (*Bilbo and Schwarz, 2009*; *Tremblay et al., 2011*; *Miyamoto et al., 2016*; *Li et al., 2019*). Although microglia were initially thought to remain quiescent until activation by neuroinflammation, in vivo imaging experiments demonstrated continual activity of their cellular processes, which actively survey their local environment (*Nakajima and Kohsaka, 2001*; *Town et al., 2005*; *Wake et al., 2009*; *Stowell et al., 2018*), including direct contact with axons and dendrites (*Schafer et al., 2013*). In addition to impacting neuronal number and initial formation of neural circuits through effects on axon targeting, microglia are thought to play an important role in synaptic refinement through selective elimination of synapses, a process termed synaptic pruning (*Paolicelli et al., 2011*; *Schafer et al., 2012*; *Hong et al., 2016*). Thus far, the majority of developmental microglia studies have focused on their role in cortical or hippocampal circuits. However, transcriptional profiling suggests a great deal of regional and temporal variation in microglial cell type and activity (*Hammond et al., 2019*; *Masuda et al., 2020*; *Young et al., 2021*), and the effects of dietary interventions have largely focused on the ARH. Here, we used Iba1 immunostaining to visualize and measure microglial morphology in regions known to receive AgRP inputs. Morphological parameters of microglia were quantified in the PVH

and ARH nuclei of the hypothalamus, as well as the bed nuclei of the stria terminalis (BST), a major limbic target of AgRP neurons, in offspring exposed to MHFD-L and compared with offspring that were raised on a normal chow diet (NCD). We also used genetically-targeted axonal labeling of AgRP neurons to directly visualize cellular interactions between microglia and labeled AgRP terminals in the PVH and ARH to determine if MHFD-L stimulates synaptic pruning in these regions. The results demonstrate regionally-specific changes to microglia in the PVH of MHFD-L offspring that are temporally restricted to the period when AgRP neurons innervate the PVH. In addition, the axon labeling experiments confirm a significant decrease in AgRP innervation of the PVH in MHFD-L offspring, and for the first time provide direct evidence of microglial-mediated synaptic pruning of AgRP terminals in the hypothalamus. Microglial depletion experiments determined that the significant decrease in AgRP innervation of the PVH observed in MHFD-L offspring requires normal densities of microglia, and that microglia are required for the weight gain seen in offspring of MHFD-L dams at weaning. However, we did not detect a significant effect of MHFD-L on the degree of synaptic pruning in the PVH, suggesting an alternative microglial signaling mechanism yet to be identified.

## Results

### Microglia exhibit morphological changes in the PVH in response to MHFD-L during postnatal development

To assess the impact of MHFD-L on microglia in the brains of postnatal mice, we used Iba1 immunohistochemistry and confocal microscopy to visualize the distribution and morphology of microglial cells in the PVH. Discrete regions of interest were imaged and a 3D modeling analysis pipeline was used to measure structural changes in microglia (*Figure 1Ai–iii*). We found that in the PVH, the overall size of microglia was significantly enhanced in offspring exposed to MHFD-L than those raised on NCD at P16 (*Figure 1B and C*). The increased size of microglia observed in MHFD-L offspring is demonstrated through a 44% increase in the complexity of microglial process branching, as determined by Sholl analysis (*Figure 1F*), as well as an 87% increase in microglial process length (*Figure 1G*). Additionally, the volume of microglial cells (volume measurements include cell body and processes; *Figure 1I*), and the spatial territory they occupy (*Figure 1H*), was nearly doubled in MHFD-L offspring, while the density of microglia between the dietary groups remained unaltered at P16 (*Figure 1J*). The changes in microglia between NCD and MHFD-L mice at P16 (*Figure 1B and C*) appeared to be transient, because by P30 (*Figure 1D and E*), we did not detect changes in microglial size (process complexity, *Figure 1F*; and length, *Figure 1G*), microglia cell territory (*Figure 1H*), microglial cell volume (*Figure 1I*), or density of microglial cells (*Figure 1J*) between MHFD-L and NCD offspring. The density of microglia in the PVH was reduced from P16 to P30, although this was independent of dietary treatment (*Figure 1J*). We also used Cre-dependent targeting of synaptophysin-tdTomato to axons of AgRP neurons to assess the density of AgRP terminals in PVH regions of interest. The results confirmed that AgRP terminal density is significantly lower in the brains of MHFD-L offspring compared with NCD controls at both P16 and P30 (*Figure 1K*). Taken together, these results suggest that exposure to MHFD-L causes changes to the morphology of microglia that are consistent with enhanced activity and surveillance of their immediate microenvironment. Furthermore, the effects of MHFD-L on microglial morphology in the PVH correspond to an accompanying decrease in the density of AgRP inputs to the PVH.

### Microglia do not exhibit morphological changes in the ARH or BST in response to MHFD-L exposure during postnatal development

The ARH houses the cell bodies of AgRP neurons, and their number is established primarily during prenatal development (*Ishii and Bouret, 2012*). We evaluated microglia morphology in the ARH of postnatal mice by using the same 3D modeling pipeline shown in *Figure 1*. In contrast to our findings in the PVH, microglial size and density in the ARH were not significantly different in NCD and MHFD-L offspring at P16 (*Figure 2A and B*; quantified in 2E-I). By P30 (*Figure 2C and D*) there was a 67% increase in process length (*Figure 2F*) of NCD mice compared to their P16 counterparts, but no significant differences in microglial morphology in the ARH were detected between the dietary treatment groups (*Figure 2A–G*). The volume of individual microglial cells (*Figure 2H*), as well as the spatial territory they occupy (*Figure 2G*), increased from P16 to P30, while the total number of microglial

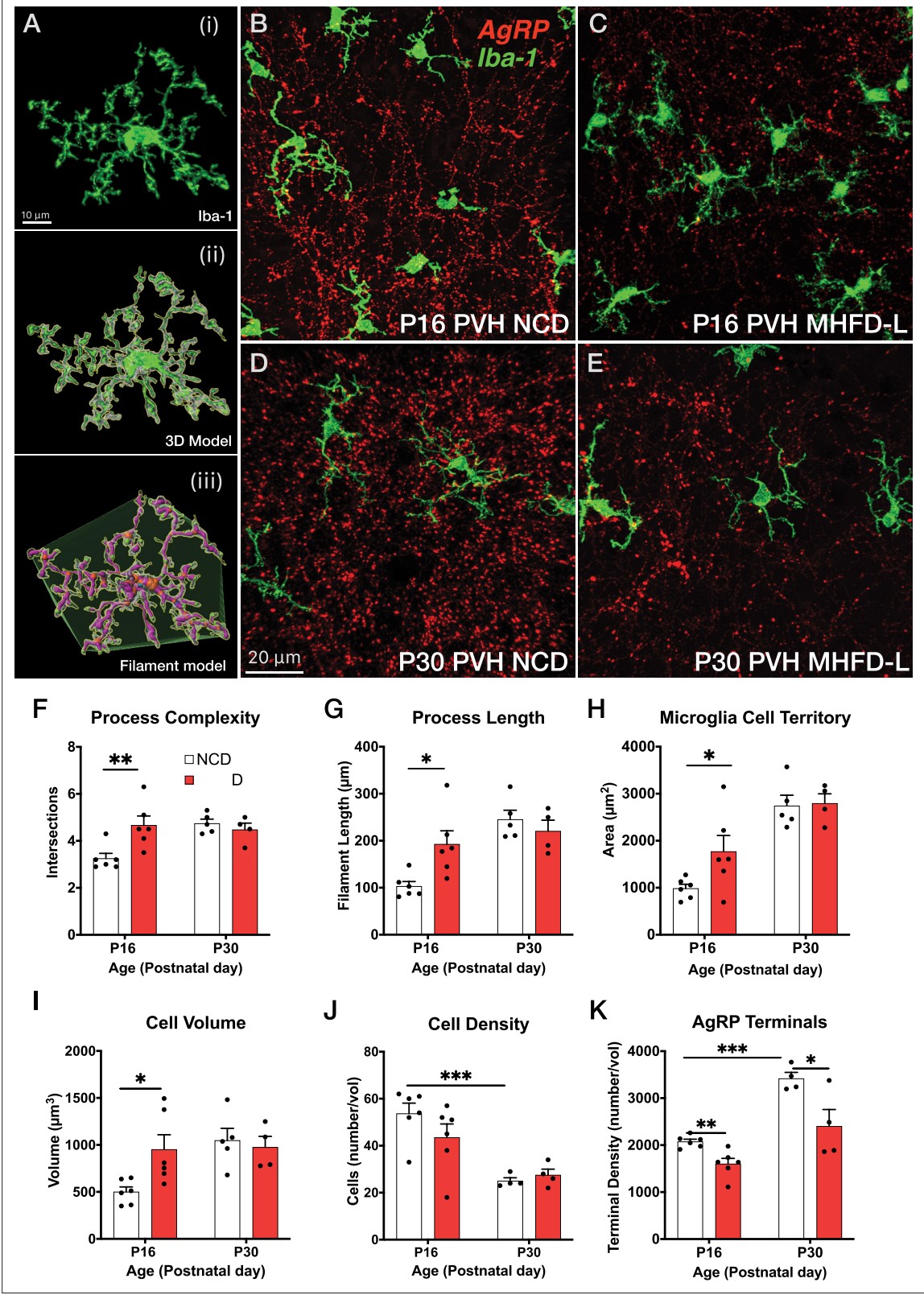

**Figure 1.** MHFD-L: Microglial morphology in the PVH. (**A**) Image analysis pipeline. Fluorescence image of an Iba1-immunostained microglial cell in the PVH (Ai). Confocal images through labeled cells were used to generate 3D reconstructions (Aii), which were then used to create 3D models of microglial cells by using the Filaments tool in Imaris. Polyhedrons were generated around each cell using the Convex Hull function of Imaris to estimate the total tissue 'territory' occupied by the microglial cell (Aiii). (**B–E**) Images of microglial cells (green) and labeled AgRP terminals (red) in the PVH of mice at P16

*Figure 1 continued on next page*

*Figure 1 continued*

(**B, C**) or P30 (**D, E**) that were raised on NCD (**B, D**) or MHFD-L (**C, E**). Graphical comparisons between groups to show that MHFD-L increased microglial ramification complexity (**F**), process length (**G**), microglial cell territory (**H**), and cell volume (**I**) at P16. The density of microglia in the PVH decreased between P16 and P30, irrespective of diet (**J**). Density of Agouti-related peptide (AgRP) terminals were decreased in MHFD-L offspring at both P16 and P30 (**K**). Bars represent the mean ± SEM and each point represents one animal. *p<0.05, **p<0.005. Abbreviations: MHFD-L, maternal HFD during lactation; NCD, normal chow diet; PVH, paraventricular nucleus of the hypothalamus.

cells was reduced by nearly half (*Figure 2I*). We also measured numbers of AgRP neuronal cell bodies in the ARH in brains derived from NCD and MHFD-L offspring and confirmed that the number of AgRP neurons in the ARH is also resistant to MHFD-L exposure at both P16 and P30 (*Figure 2J*). In addition, we evaluated microglia and AgRP terminals in the anterolateral part of the BST (*Figure 3*), an extrahypothalamic target of AgRP neurons innervated during the lactational period (*Bouret et al., 2004a*; *Cansell et al., 2012*; *Barbier et al., 2021*). As was found for the ARH, neither the size nor number of microglia in the BST were significantly different between NCD and MHFD-L offspring at P16 or P30 (*Figure 3A–D*; quantified in 3E-I). Similarly, the density of AgRP terminals in the same region of interest was not affected by MHFD-L exposure (*Figure 3J*), although by P30 the number of AgRP terminals in the BST increased by 61% compared to their P16 counterparts (*Figure 3J*). Taken together, these data suggest that microglia in the ARH and BST are resistant to the morphological changes that occur in the PVH of MHFD-L offspring, suggesting a notable degree of spatial specificity in the role of hypothalamic microglia during postnatal development.

## Microglia are required for changes in AgRP terminal density in PVH and body weight associated with MHFD-L exposure

To determine if microglia are required for the observed changes in AgRP inputs to the PVH of MHFD-L offspring, the colony-stimulating factor 1 receptor (CSF1R) inhibitor PLX5622 was administered from P4-P21 via daily intraperitoneal (i.p.) injection (*Figure 4A*). These postnatal treatments resulted in a significant decrease in microglia detected in the hypothalamus at P55 relative to age-matched controls (*Figure 4B and C*; quantified in 4 H). Notably, the PLX5622 treatments blocked the reduction in AgRP fiber density observed in the medial dorsal parvicellular part of the PVH (PVHmpd) of vehicle-treated MHFD-L offspring to a level that was comparable to that of NCD offspring (*Figure 4D–G*; quantified in 4I). In contrast, MHFD-L exposure did not affect the density of AgRP fibers in the lateral posterior magnocellular compartment of the PVH (PVHpml) and no significant difference in AgRP fiber density was detected between NCD and MHFD-L offspring treated with either PLX5622 or vehicle (*Figure 4D–G*; quantified in 4 K), suggesting target specificity for the microglial-mediated effects on development of AgRP inputs to the PVHmpd.

Depletion of microglia also appeared to protect against the increase in body weight normally observed in MHFD-L mice. In keeping with previously published results (*Vogt et al., 2014*), the body weight of vehicle-treated MHFD-L animals was significantly greater (24%), compared with that of NCD animals at weaning (*Figure 4J*). However, MHFD-L mice treated with PLX5622 during lactation exhibited significantly lower weights at weaning compared to those of vehicle-treated MHFD-L animals (*Figure 4J*). Taken together, these findings suggest that microglia mediate target-specific effects of MHFD-L exposure on the innervation of the PVH by AgRP neurons, and that microglia play a role in mediating the effects of MHFD-L on body weight.

## Engulfment of AgRP terminals by microglia in the PVH and the ARH

Microglia are thought to impact the development of neuronal connections through an active engulfment mechanism and the lysosomal-associated membrane protein CD68 has been implicated in this process. Here, we used immunohistochemistry to visualize the presence of Iba1-labeled microglia in mice with genetically targeted labeling of AgRP terminals (*Figure 5*). Many apparent contacts between microglial processes and AgRP terminals in the PVH and ARH were observed at P16 and P30, including internalized AgRP terminals (*Figure 5A–L*). However, the extent of internalization did not appear to be influenced by MHFD-L exposure; there were no significant differences between internalized AgRP terminals in MHFD-L and NCD offspring at P16, in either the PVH or ARH (*Figure 5M and O*). Similarly, we did not detect a statistically significant difference in microglial CD68 levels in the PVH between diet groups at either P16 or P30 (*Figure 5N*). Consistent with previous reports

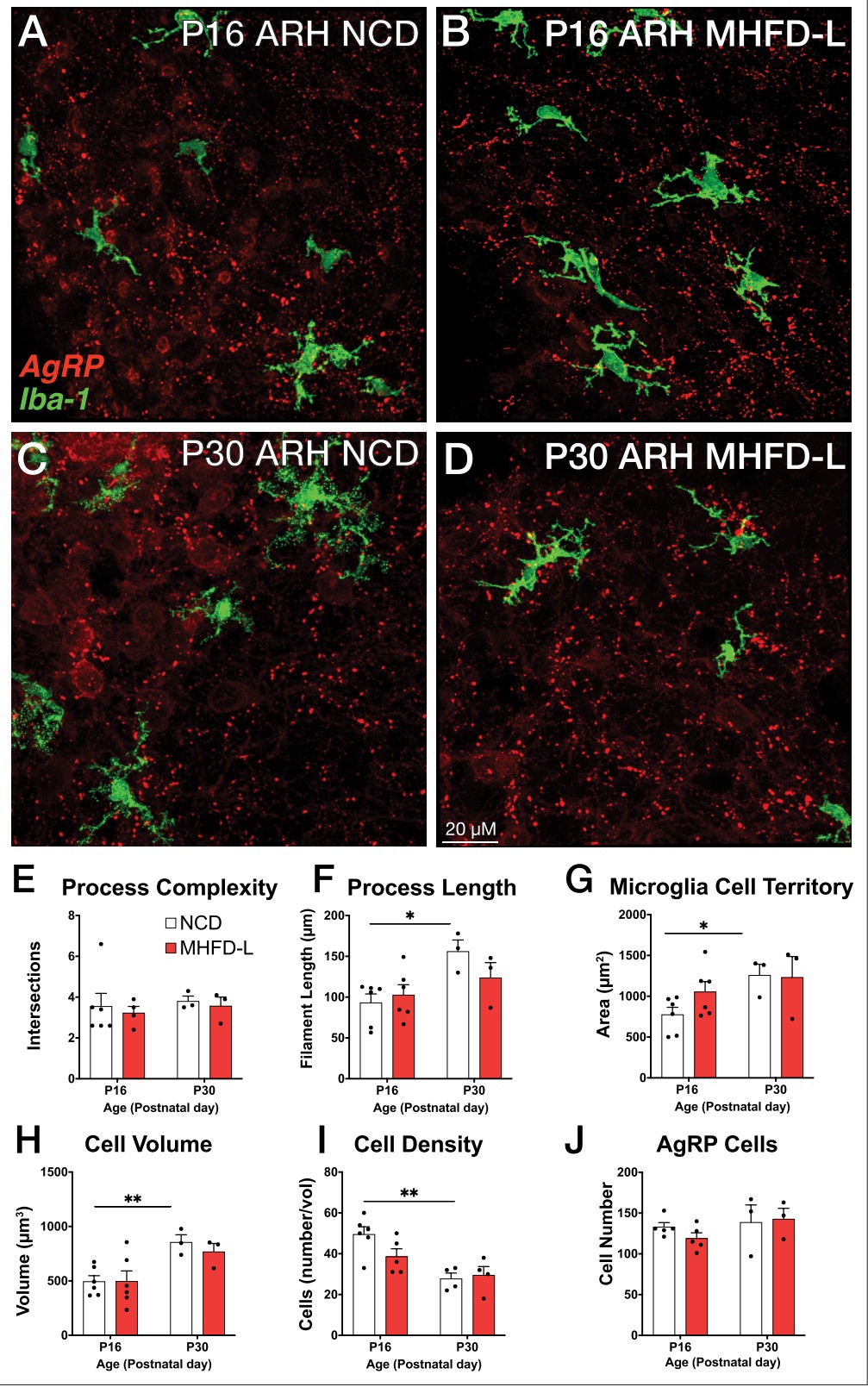

**Figure 2.** MHFD-L: Microglial morphology in the ARH. Microglial cells (green) and labeled Agouti-related peptide (AgRP) terminals (red) in the ARH of mice at P16 (**A, B**) or P30 (**C, D**) that were exposed to NCD (**A, C**) or MHFD-L (**B, D**). Graphical comparisons between groups to show that microglial ramification complexity (**E**) remained the same, regardless of age or diet. Microglial process length (**F**), cell territory (**G**), and cell volume (**H**) increased

*Figure 2 continued on next page*

*Figure 2 continued*

between P16 and P30, but were not changed as a result of diet. The density of microglia in the ARH decreased between P16 and P30, irrespective of diet (**I**). There were no apparent changes in the numbers of AgRP neurons (**J**). Bars represent the mean ± SEM and each point represents one animal. *p<0.05, **p<0.005. Abbreviations: ARH, arcuate nucleus of the hypothalamus; MHFD-L, maternal HFD during lactation; NCD, normal chow diet.

(*Wong et al., 2005*; *Hart et al., 2012*), the density of CD68 labeled profiles nearly doubled in the PVH between P16 and P30 (*Figure 5N*), as microglia become more phagocytic with age. In the ARH, CD68 staining also increased between P16 and P30 (*Figure 5P*), supporting the notion that microglia increase their phagocytic capacity with age. Nevertheless, our analysis demonstrates that microglia interact directly with AgRP terminals, with clear evidence of engulfment. MHFD-L exposure does not appear to promote microglia-mediated engulfment, at least not in the specific PVH and ARH domains examined.

## Discussion

It is well established that microglia are responsive to HFD exposure in adult rodents (*Thaler et al., 2012*; *Morari et al., 2014*; *Valdearcos et al., 2014*; *Baufeld et al., 2016*; *Valdearcos et al., 2017*) and multiple lines of evidence support an important role for microglia in mediating key aspects of neural circuit development (*Checchin et al., 2006*; *Hoshiko et al., 2012*; *Li et al., 2012*; *Hagemeyer et al., 2017*). Here, we demonstrate that microglia are required for significant elevations in body weight that emerge from postnatal exposure to HFD and are associated with a sustained decrease in the density of afferents from AgRP neurons to the PVH in offspring. Exposure to MHFD-L caused distinct morphological changes to microglia that may be consistent with enhanced activity, which were observed in the PVH, but not the ARH or BST. Moreover, the morphological changes to microglia observed appear to be primarily limited to the critical period for the development of AgRP inputs to PVH neurons in MHFD-L offspring. Although our results demonstrate that microglia engage in engulfment of AgRP terminals in the PVH during development, synaptic pruning by microglia does not appear to represent the cellular mechanism mediating the effects of MHFD-L exposure on innervation of the PVH by AgRP neurons.

### A role for microglia in mediating body weight changes observed in MHFD-L offspring

Maternal HFD exposure during perinatal development leads to increased body weight, fat content, and susceptibility to diet-induced obesity in offspring (*Samuelsson et al., 2008*; *Tamashiro et al., 2009*; *Masuyama and Hiramatsu, 2014*). The postnatal period, which corresponds to lactation, and when mice derive their nutrition primarily from milk, is especially sensitive to nutritional environment and exposure to HFD that is exclusive to this period plays a particularly dominant role in specifying metabolic phenotype later in life (*Chen et al., 2008*; *Sun et al., 2024*; *Vogt et al., 2014*). In the present study, global depletion of microglia with the CSF1R inhibitor PLX5622 blocked the ability of MHFD-L to increase body weight in mice by weaning at P21, indicating microglia may mediate metabolic changes caused by MHFD-L exposure. Consistent with this, microglial depletion with PLX5622 in adult mice mitigates the effects of HFD exposure (*Rosin and Kurrasch, 2019*), and our results indicate that microglia may function similarly in offspring during postnatal life to effect changes in body weight, even if the maternal HFD exposure is restricted to the lactation period. Further studies are required to define the long-term metabolic profile resulting from developmental microglial manipulations. However, given the abundant literature on the sustained impact of MHFD-L on metabolic phenotype, enduring disruptions are likely. It should be noted that PLX5622 treatment is not spatially limited to the PVH or ARH, leaving open the possibility that the effects of microglial depletion on body weight occur outside of these nuclei, or are due to collective activation of microglia in multiple components of feeding circuitry (*Green et al., 2020*). Localization of the specific site of action for microglial specification of mature body weight during development will require utilization of specific markers for hypothalamic microglia that account for regional and phenotypic heterogeneity, perhaps through intersectional genetic methods and combinatorial pharmacology (*Hammond et al., 2019*; *Kim et al., 2021*).

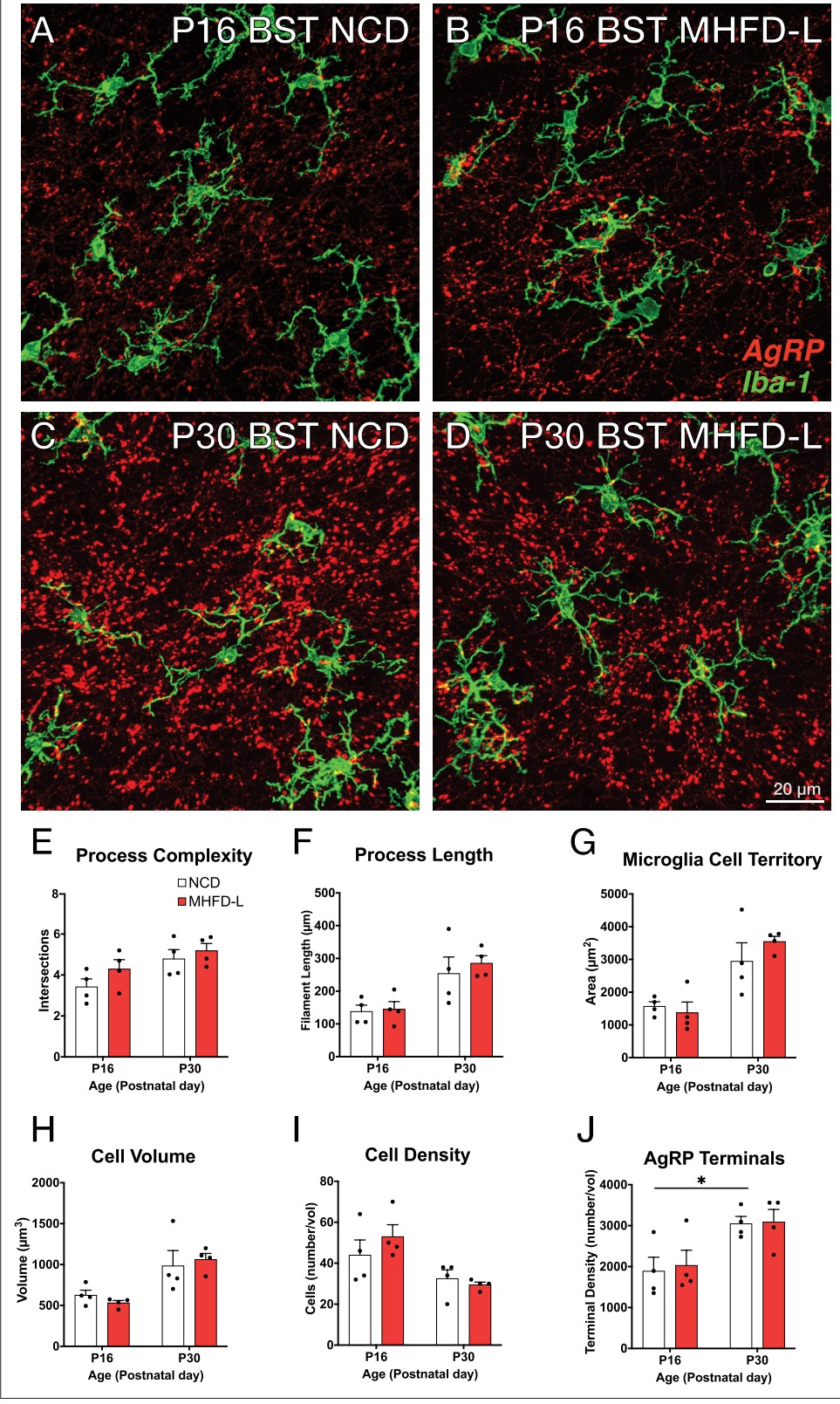

**Figure 3.** MHFD-L: Microglial morphology in the BST. Microglial cells (green) and labeled Agouti-related peptide (AgRP) terminals (red) in the BST of mice at P16 (**A, B**) or P30 (**C, D**) that were exposed to NCD (**A, C**) or MHFD-L (**B, D**). Graphical comparisons between groups to show that microglial ramification complexity (**E**), process length (**F**), cell territory (**G**), and cell volume (**H**) did not significantly change between P16 and P30, nor were they changed

*Figure 3 continued on next page*

*Figure 3 continued*

as a result of diet. The density of microglia in the BST decreased between P16 and P30, irrespective of diet (**I**). The density of AgRP terminals increased between P16 and P30, but there was no effect of maternal diet (**J**). Bars represent the mean ± SEM and each point represents one animal. *Pp<0.05. Abbreviations: BST, bed nucleus of the stria terminalis; MHFD-L, maternal HFD during lactation; NCD, normal chow diet.

## MHFD-L induces spatially limited changes in microglial morphology

Morphological changes in microglia have been reported in response to a variety of environmental exposures. In the hypothalamus, adult mice fed a HFD show both proliferation and changes in microglial process length and complexity (*Thaler et al., 2012*; *Valdearcos et al., 2017*). In our studies, MHFD-L exposure caused a marked increase in the overall size of microglia in the PVH that is related to increases in both the length and branching complexity of immunolabeled Iba1 cellular processes. This increase in the territory occupied by microglia in the PVH was not accompanied by

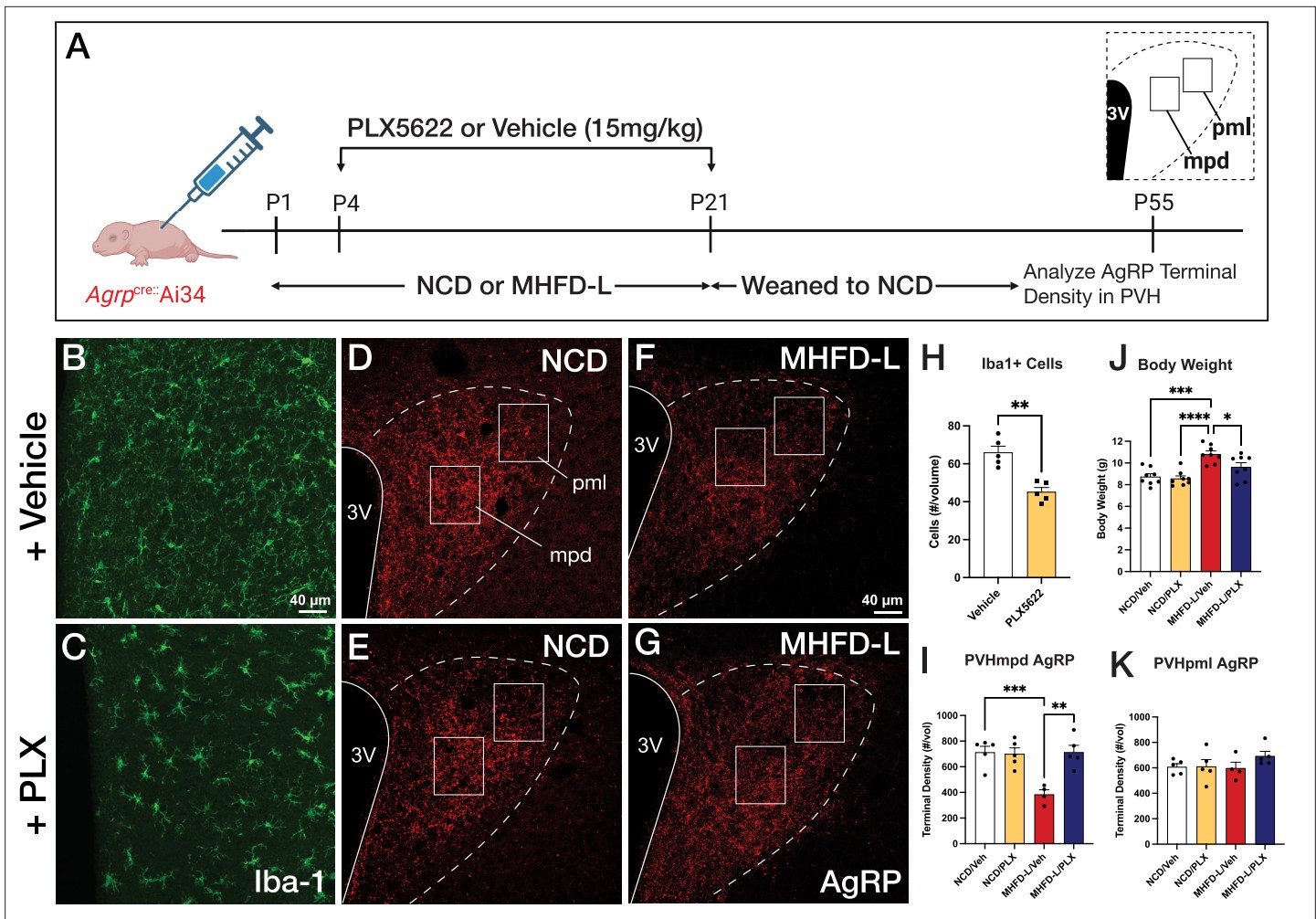

**Figure 4.** Microglial depletion during lactation period. Schematic of MHFD-L exposure and PLX5622 treatment experimental design (**A**). Microglial cells (green) in the PVH of adult mice (P55) treated during lactation with vehicle (**B**) or PLX5622 (**C**). Images of labeled AgRP terminals (red) to illustrate the density of labeling in distinct compartments of the PVH (white boxes denote locations of ROIs) of normal chow diet (NCD) offspring (**D ,E**) and MHFD-L offspring (**F, G**). Graphical comparisons to illustrate the effects of postnatal PLX5622 treatments on microglia density in the PVH (**H**), body weight (**J**), and the density of AgRP terminals in the PVHmpd (**I**) and PVHpml (**J**). Bars represent the mean ± SEM and each point represents one animal. Unpaired t-test was used to compare cell number in 4 H; two-way ANOVA was used to test for differences in group means, followed by Tukey's multiple comparisons posthoc test to identify specific group differences in 4I-K. Pp-values less than 0.05 were considered significant; *p<0.05, **p<0.005, ***p<0.0005. Abbreviations: AgRP, agouti-related peptide; CSF1R, Colony-stimulating factor 1 receptor; MHFD-L, maternal HFD during lactation; PVH, paraventricular nucleus of the hypothalamus; mpd, medial parvocellular compartment of the PVH; pml, posterior magnocellular compartment of the PVH.

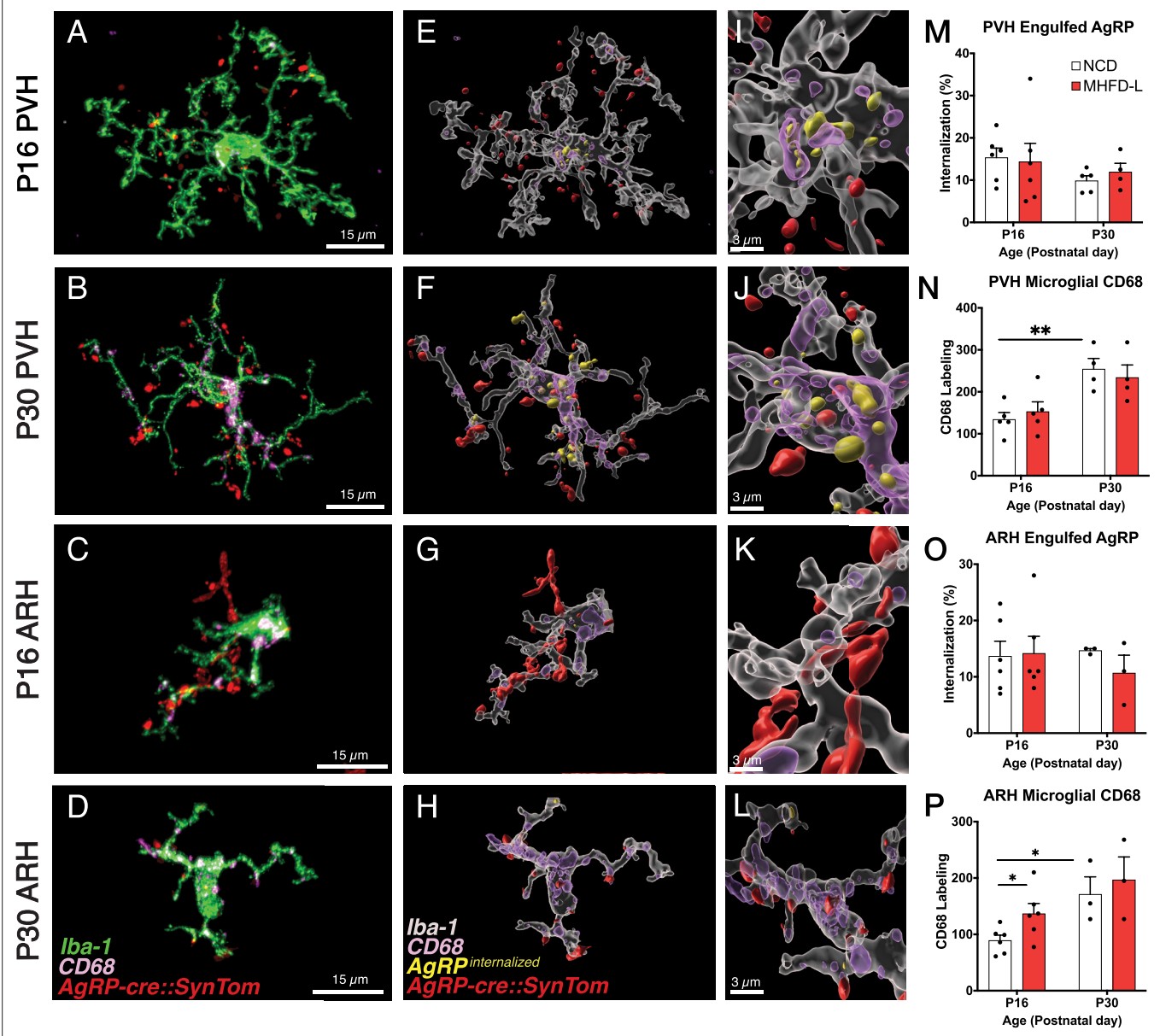

**Figure 5.** Microglial interaction with AgRP axon terminals in the PVH and arcuate nucleus of the hypothalamus (ARH). (**A–D**) Representative images of microglial cells (green), labeled AgRP terminals (red), and CD68 (lysosomal associated membrane protein and phagocytic capacity marker, pink) that compare their cellular relationships in the PVH (**A, B**) and ARH (**C, D**) at P16 and P30. (**E–H**) Digital 3D reconstructions of cells shown in (**A–D**) after application of filaments tool to visualize internalized AgRP terminals. (**I–L**) Cells shown in E-H after application of digital zoom to more clearly illustrate engulfment of labeled AgRP terminals by microglia and location of CD-68 labeled profiles. (**M–P**) Graphical comparisons between groups to illustrate the effects of age and MHFD-L exposure on CD68 expression and AgRP terminal engulfment. Bars represent the mean ± SEM and each point represents one animal. *Pp<0.05, **Pp<0.005. Abbreviations: AgRP, agouti-related peptide; ARH, arcuate nucleus or the hypothalamus; CD68, Cluster of Differentiation 68; MHFD-L, maternal HFD during lactation; PVH, paraventricular nucleus of the hypothalamus.

an increase in microglial number, nor were numbers of microglia affected in the PVH by MHFD-L exposure. However, in contrast to the PVH, changes in microglial morphology were not observed in the ARH in response to MHFD-L, although we did observe an overall increase in process length in the ARH between P16 and P30 of both MHFD-L and NCD mice. This finding is consistent with previously published reports on microglial maturation (*Sun et al., 2024*). The spatially restricted enhancement of microglial activation in the PVH resulting from MHFD-L exposure appears to contribute to an expansion of parenchymal territory surveilled by PVH microglia, as reflected in the volume measurements accomplished with geometrical modeling of process length and complexity.

This interpretation is supported by in vitro and in vivo observations of enhanced process extension and increased neuronal interactions resulting from inflammatory activation of microglia (*Wake et al., 2009*; *Schafer et al., 2013*; *Dissing-Olesen et al., 2014*; *Stowell et al., 2018*; *Bolton et al., 2022*).

As in the ARH, we did not observe comparable changes in microglial morphology in the BST, an extrahypothalamic target of AgRP neurons innervated during the lactation period (*Cansell et al., 2012*; *Barbier et al., 2021*). These observations underscore the remarkable molecular heterogeneity of microglial phenotypes that appear to occupy various hypothalamic niches during development, and may have equally diverse developmental roles and responses to environmental signals (*Bilbo and Schwarz, 2012*; *Frost and Schafer, 2016*; *Li and Barres, 2018*; *Ngozi and Bolton, 2022*). Moreover, the observed morphological changes in the PVH caused by MHFD-L appear to be transient as there are no significant differences in microglial processes by P30, and the density of microglial cells in the PVH was significantly reduced in the older mice, suggesting a decline in overall activity. That the observed changes in microglial morphology occur within the critical period for the development of AgRP projections to the PVH suggests a possible role for microglia linking nutrition with specification of axonal targeting (*Kamitakahara et al., 2018*).

## Microglia mediate impaired innervation of the PVH by AgRP neurons

AgRP neuronal projections develop primarily during the first 2 wk of life, which corresponds to a critical period for the neurotrophic action of leptin on axonal outgrowth and targeting of AgRP inputs to distinct components of the PVH (*Bouret et al., 2004a*; *Elson and Simerly, 2015*). Exposure to MHFD-L during this critical developmental window permanently impairs AgRP projections to the PVH, DMH, and LH (*Vogt et al., 2014*) that are associated with increased body weight in adulthood. Here, we confirm that MHFD-L impairs innervation of the PVH, and this defect was apparent by P16. The PVH is innervated by AgRP neurons between P8 and P10 (*Bouret et al., 2004a*). We found that depletion of microglia with PLX5622 between P4 and P21, a period that aligns not only with AgRP innervation of the PVH but also with maximum changes in microglial morphology, blocked the effects of MHFD-L on AgRP terminals in the PVH. However, this partial rescue of innervation appeared to be limited to the PVHmpd, suggesting that there is a regional specialization in the activity of microglia in the PVH. We did not observe a change in AgRP innervation of the BST in MHFD-L offspring that mirrored the changes seen in the PVH of the same animals, further supporting the conclusion that microglia display at least some degree of spatial heterogeneity in mediating site-specific alterations in AgRP axon targeting.

Microglia are known to impact a variety of developmental events, including alterations in cell number through programmed cell death or neurogenesis, as well as axonal targeting and remodeling of neural circuits (*Frost and Schafer, 2016*; *Li and Barres, 2018*). Because PLX5622 causes a decrease in microglia in the ARH as well as in the PVH, our microglial depletion studies do not eliminate the possibility that microglia may act locally on AgRP neurons to affect axonal growth. It is unlikely that the impaired innervation of the PVH observed in MHFD-L offspring is due to a reduction in the number of AgRP neurons in the ARH (*Vogt et al., 2014*; *Valdearcos et al., 2017*). *Sun et al., 2024* reported that microglial depletion during postnatal life increases numbers of NPY neurons, as well as enhances local densities of AgRP fibers in the ARH, possibly through enhanced formation of perineuronal nets. Depletion of microglia during gestation causes a significant decrease in the number of proopiomelanocortin (POMC) neurons in the ARH and leads to an acceleration of weight gain (*Rosin and Kurrasch, 2019*), consistent with neurogenesis of ARH neurons occurring in mid-gestation and increased susceptibility to nutritional impacts during embryonic life (*Ishii and Bouret, 2012*; *Elson and Simerly, 2015*). Interestingly, genetic deletion of leptin receptors from myeloid cells reduced numbers of POMC neurons in the ARH, suppressed POMC innervation of the PVH, and decreased microglial process complexity (*Gao et al., 2018*). Taken together, these results suggest that leptin signaling in microglia may act at the level of the ARH to promote outgrowth of AgRP projections to the PVH, while MHFD-L activates microglia in the PVH to specify patterns of AgRP afferents that are not only regionally specific, but also target discrete subdomains of the PVH. However, whether microglia inhibit synaptogenesis or are involved in synaptic refinement through an alternative regressive mechanism will require further investigation.

## Microglia in PVH Participate in Synaptic Pruning

Microglia have been proposed as mediators of synaptic pruning, a process whereby synapses that form early in development are eliminated as others are strengthened and maintained (*Katz and Shatz, 1996*; *Sanes and Lichtman, 1999*; *Frost and Schafer, 2016*; *Li and Barres, 2018*). Although this process has been studied most extensively in somatosensory cortex (*Miyamoto et al., 2016*), hippocampus (*Paolicelli et al., 2011*; *Wang et al., 2020*), and the visual system (*Tremblay et al., 2010*; *Schafer et al., 2012*), there is evidence for involvement of microglia in synaptic pruning of immuno-labeled glutamatergic terminals associated with corticotropin-releasing hormone neurons in the PVH (*Bolton et al., 2022*). In the present study, we used genetically targeted axonal labeling to provide evidence that microglia in the PVH participate in synaptic pruning of AgRP synapses during the critical period for PVH innervation, and when PVH microglia exhibit high levels of process extension. The lysosomal marker CD68 was colocalized with internalized AgRP terminals in PVH microglia, and although elevated at P30, there were no differences between offspring of MHFD-L and NCD dams. Additionally, we did not find a significant difference in the density of engulfed AgRP terminals in the PVH of MHFD-L offspring at either P16 or P30. However, enhanced engulfment of AgRP terminals in MHFD-L offspring may occur at a later point in development not assessed in this study. It is also possible that the synaptophysin-tdTomato axonal label may have been lost from pruned synapses during engulfment. A more probable interpretation is that PVH microglia participate in synaptic refinement through other cellular mechanisms, including microglial release of secreted factors such as the interleukin IL-6 (*Kim et al., 2024*) or microglial-derived BDNF (*Parkhurst et al., 2013*). Future studies that include cell type-specific manipulations of microglial signaling and live cell imaging may clarify these potential developmental mechanisms.

## Conclusions

MHFD-L causes elevated levels of saturated carbohydrates and fats in milk (*Gorski et al., 2006*; *Vogt et al., 2014*; *Calvo-Lerma et al., 2022*). The resulting overnutrition resulting from exposure to this enhanced diet is thought to underlie the propensity towards obesity observed in offspring later in life (*Skowronski et al., 2024*). Although microglia are likely mediators of multiple neurobiological events influencing how hypothalamic circuits function during regulation of energy balance, the precise signaling mechanisms remain ill-defined. There may be common molecular mechanisms underlying the effects of HFD exposure on microglial activation in adults and those occurring during postnatal development, but how these signaling events exert a lasting impact on the organization and function of feeding circuitry has not been defined. The results presented here demonstrate an important role for microglia on sculpting the density of inputs from AgRP neurons to the PVH that is not only spatially restricted, but also aligned temporally with synaptogenesis in the PVH. Furthermore, PVH microglia clearly interact directly with AgRP afferent axons during this critical period and may be refined through engulfment by microglia. However, synaptic pruning through engulfment does not appear to be sufficient to affect the significant reduction in AgRP innervation of PVH neurons observed following MHFD-L exposure, suggesting involvement of additional microglial signaling mechanisms that are not only important for specifying patterns of innervation of the PVH by AgRP neurons, but may also contribute more broadly to developmental programming of metabolic phenotype.

# Materials and methods

**Key resources table**

| Reagent type (species) or resource | Designation | Source or reference | Identifiers | Additional information |
|---|---|---|---|---|
| Genetic reagent (*M. musculus*) | *Agrp*$^{tm1(cre)Lowl}$/J | Jackson Laboratory | Stock #: 012899 RRID:IMSR_JAX:012899 | MGI ID: J:140858 |
| Genetic reagent (*M. musculus*) | Ai34(RCL-Syp/tdT)-D (B6;129S-*Gt(ROSA)26Sor*$^{tm34.1(CAG-Syp/tdTomato)/Hze}$/J) | Jackson Laboratory | Stock #: 012570 RRID:IMSR_JAX:012570 | MGI ID: J:170755 |
| Antibody | Rabbit polyclonal anti-Iba1 | FUJIFILM Wako | Cat. #: 019–19741 RRID:AB_839504 | IHC (1:2000) |

*Continued on next page*

*Continued*

| Reagent type (species) or resource | Designation | Source or reference | Identifiers | Additional information |
|---|---|---|---|---|
| Antibody | Rat monoclonal anti-CD68 [FA-11] | Abcam | Cat. #: ab53444 RRID:AB_869007 | IHC (1:500) |
| Antibody | Donkey polyclonal anti-rabbit Alexa Fluor 488 | ThermoFisher Scientific | Cat. #: A32790 RRID:AB_2762833 | IHC (1:500) |
| Antibody | Donkey polyclonal anti-rat Alexa Fluor 647 | ThermoFisher Scientific | Cat. #: A48272 RRID:AB_2893138 | IHC (1:500) |
| Chemical compound | PLX5622 hemifumarate, CSF1R inhibitor | MedChemExpress | Cat. #: HY114153A | |
| Software, Algorithm | Imaris | Bitplane | V9.5 | |
| Software, Algorithm | GraphPad Prism | Prism | Prism 10 | |

## Animals

All animal care and experimental procedures were performed in accordance with the guidelines of the National Institutes of Health and the Institutional Care and Use Committee of Vanderbilt University, protocols #M1700113-00 and #M2300065-00. Mice were housed at 22 °C on a 12:12 hr light:dark cycle provided ad libitum access to NCD (PicoLab Rodent Diet 20 #5053) and water unless otherwise specified. AgRP-Cre mice (*Agrp*$^{tm1(cre)Lowl}$/J; stock number: 01289) and mice expressing the Cre-dependent fluorescent reporter synaptophysin-tdTomato (RCL-Syp/tdT)-D (B6;129S-*Gt(ROSA)26Sor*$^{tm34.1(CAG-Syp/td-Tomato)/Hze}$/J; stock number: 012570) were obtained from the Jackson Laboratory (Bar Harbor, ME) and maintained in our colony at Vanderbilt University. To visualize AgRP inputs, *Agrp*-Cre mice were crossed with Ai34D mice to generate Agrp-Cre::Ai34D mice, as described previously (*Biddinger et al., 2020*).

To generate offspring of dams exposed to HFD during lactation (MHFD-L), mice had ad libitum access to NCD (PicoLab Rodent Diet 20 #5053: 25% protein; 62% carbohydrates; 13% fat; 4 kcal/g energy density) prior to and during mating. On the first postnatal day (P1) all litters were adjusted to seven pups to normalize nutrition and dams were switched to either HFD (Research Diets D12451: 20% protein; 35% carbohydrate; 45% fat; 4.7 kcal/g energy density) or kept on the same NCD. The dams remained on either HFD or NCD throughout lactation, and offspring were weaned onto the same normal chow diet that the dams had received prior to MHFD-L treatment, regardless of lactation dietary condition, and remained on NCD until they were processed for perfusion.

## PLX5622 microglia depletion

To reduce microglia during postnatal development, mouse pups were treated daily from P4 to P21 with either PLX5622, a colony-stimulating factor 1 receptor (CSF1R) inhibitor or DMSO vehicle via intraperitoneal injection (*Riquier and Sollars, 2020*). Briefly, PLX5622 hemifurate solid (Cat. #HY114153A MedChemExpress, Monmouth Junction, NJ, USA) was suspended in DMSO at a concentration of 172 mg/ml. The injection working solution was prepared to include 20% Kolliphor RH40 diluted in PBS, which resulted in doses with a 6.5 mg/ml PLX5622 concentration and injection concentration of 15 mg/kg.

## Immunohistochemistry

Mice were perfused at P16 and P30 and processed for immunofluorescence by using primary antibodies to Iba1 (1:2000; FUJIFILM Wako, Osaka, Japan) to visualize microglia and CD68 (1:500; Abcam, Cambridge, MA, USA) to assess phagocytic capacity of the microglia. Mice were first anesthetized with tribromoethanol (TBE) and then perfused transcardially with cold 0.9% saline, followed by cold fixative (4% paraformaldehyde in borate buffer, pH 9.5) for 10 min. After fixative perfusion, brains were removed from the skull and postfixed in the same fixative overnight. The next day, the tissue was transferred to 20% sucrose for cryoprotection overnight. A freezing-stage sliding microtome was used to collect 30 µm-thick coronal sections and free-floating tissue sections were stored in cryoprotectant solution at –20 °C until further processing. To prepare tissue

for immunohistochemical processing, brain sections were removed from cryoprotectant and rinsed several times in 0.02 M KPBS. Free-floating sections were incubated in blocking buffer containing 2% normal donkey serum and 0.3% Triton-X 100 in 0.02 M KPBS overnight at 4 °C. Tissue sections were incubated in the same blocking buffer with primary antibodies for 48 hr at 4 °C. Following primary antibody incubation, sections were rinsed several times in KPBS, and then incubated in the appropriate species-specific fluorophore-conjugated Alexa-Fluor secondary antibodies for 1 hr at RT. Tissue sections were again rinsed several times with KPBS, and mounted onto charged microscope slides and coverslipped using ProLong antifade mounting medium (Life Technologies, Carlsbad, CA, USA).

## Image acquisition and analysis

Sections through the PVH, ARH, and BST were identified and morphological features of the nuclei were visualized with Hoescht 33342, which were then used to define matching regions of interest (ROI) for quantitative analysis carried out by a user blind to treatment group. Confocal image stacks were collected using a laser scanning confocal microscope (Zeiss LSM 800) for each ROI through the entire thickness of the region at a frequency of 0.1 µm using the 40 x objective. Imaris visualization software (Bitplane V9.5, Salisbury Cove, ME, USA) was used to create 3D reconstructions of each multichannel set of images.

Profiles of Iba1-immunolabeled microglia were segmented and skeletonized by using the Filaments tool in Imaris to quantify changes in microglia structure. Sholl analysis (*Derecki et al., 2014*), was performed on the skeletonized structures to determine the branching complexity of microglial processes. Briefly, 3D concentric spheres are drawn around each selected microglial cell and contact points between microglial processes and the spheres are counted. 3D reconstructions of the microglia were also evaluated for process length and cell volume. To estimate the 3D space occupied by each microglial cell, a polyhedron was drawn around the microglia by using the built-in Convex Hull function under Filaments to estimate the regional volume occupied by each cell analyzed.

In order to assess cellular interactions between AgRP terminals and microglia, 3D renderings of microglia and AgRP terminals were used to determine densities of contact points between AgRP terminals and microglial processes by using an Imaris MATLAB script to automate the analysis. AgRP terminals were reconstructed as 'spots' of 0.8 mm diameter (corresponding to the largest measured size) and their total number was calculated in each ROI. Briefly, the automatic detection algorithm applies a 3D Mexican hat filter using the spot size and then locates the spot centroid at the local maxima of the filtered image. The number of spots located at no more than 1 µm from the microglia surface was automatically determined as an estimate of contact points between AgRP terminals and microglia. Next, spots that were determined to be more than 0.5 µm away from internal microglial surfaces were determined to be internalized by the microglia and counted as engulfed AgRP terminals. Intracellular CD68 levels in microglia were estimated by segmentation of CD68 labeling profiles in image stacks followed by creation of 3D renderings and their volume computed. Finally, to assess the effect of MHFD-L on numbers of AgRP neurons in the ARH, AgRP neuronal cell bodies were identified by aligning synaptophysin-tdTomato labeled somal profiles with Hoescht 33342-stained nuclei and counting the number of visualized neurons manually.

## Statistical analyses

Data are presented as group mean values ± SEM, as well as individual data points. Statistical analyses were performed using GraphPad Prism software (Version 10). Unpaired t-tests were used to compare data between two groups. In the microglia depletion experiment, two-way analysis of variance (ANOVA) was used to test for differences in group means, followed by Tukey's multiple comparisons post hoc test to identify specific group differences. Differences between groups were considered statistically significant at $p < 0.05$.

## Acknowledgements

We thank the members of the Simerly Lab for comments and discussion on early versions of this manuscript and Nicholas Thomas-Low for assistance with the figures. This work was supported by NIH grants R01DK106476 (RBS) and T32DK07563 (HNM-R).

# Additional information

## Funding

| Funder | Grant reference number | Author |
|---|---|---|
| National Institute of Diabetes and Digestive and Kidney Diseases | T32DK07563 | Haley N Mendoza-Romero |
| National Institute of Diabetes and Digestive and Kidney Diseases | R01DK106476 | Richard Simerly |

The funders had no role in study design, data collection and interpretation, or the decision to submit the work for publication.

## Author contributions

Haley N Mendoza-Romero, Conceptualization, Data curation, Formal analysis, Investigation, Methodology, Writing - original draft, Writing - review and editing; Jessica E Biddinger, Formal analysis, Supervision, Validation, Investigation, Writing - original draft, Writing - review and editing; Michelle N Bedenbaugh, Writing - review and editing; Richard Simerly, Conceptualization, Resources, Supervision, Funding acquisition, Writing - original draft, Writing - review and editing

## Author ORCIDs

Haley N Mendoza-Romero ⓘ https://orcid.org/0000-0001-7290-9246
Jessica E Biddinger ⓘ https://orcid.org/0000-0001-7718-4782
Richard Simerly ⓘ https://orcid.org/0000-0001-5840-0152

## Ethics

All animal care and experimental procedures were performed in accordance with the guidelines of the National Institutes of Health and the Institutional Care and Use Committee of Vanderbilt University, protocols #M1700113-00 and #M2300065-00.

Reviewer #1 (Public review): https://doi.org/10.7554/eLife.101391.3.sa1
Reviewer #2 (Public review): https://doi.org/10.7554/eLife.101391.3.sa2
Reviewer #3 (Public review): https://doi.org/10.7554/eLife.101391.3.sa3
Author response https://doi.org/10.7554/eLife.101391.3.sa4

# Additional files

## Supplementary files

MDAR checklist

## Data availability

All data collected in this study are included in the manuscript and have been deposited at Open Science Framework: https://doi.org/10.17605/OSF.IO/J2UD3.

The following dataset was generated:

| Author(s) | Year | Dataset title | Dataset URL | Database and Identifier |
|---|---|---|---|---|
| Mendoza-Romero HN, Biddinger JE, Bedenbaugh MN, Simerly RB | 2024 | Microglia are Required for Developmental Specification of AgRP Innervation in the Hypothalamus of Offspring Exposed to Maternal High Fat Diet During Lactation | https://osf.io/j2ud3/ | Open Science Framework, 10.17605/OSF.IO/J2UD3 |

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
