## [Editor Report · eLife Assessment]

The authors provide a **valuable** contribution by documenting the role of microglia in pruning the axon terminals of AgRP neurons. The analysis of microglial axonal pruning is **solid**; however, the analysis of the effects inhibiting microglia on subsequent food consumption is not fully complete.

---

## [Referee Report · Reviewer #1 (Public review)]

Summary:

This paper shows that maternal high-fat diet during lactation changes microglia morphology in the PVN, potentially to acquire a more active state. Further, the authors reveal that PVN microglia engulf AgRP terminals in the PVN during postnatal development, a previously unrecognized behavior. A notable finding of this paper is that pharmacological reduction of microglial cells can reverse weight gain and terminal loss in the offspring under maternal high fat diet conditions, even though an increase in microglial engulfment of AgRP+ terminals was not observed, suggesting an alternative mechanism. The data support these findings, although questions remain regarding the efficacy and timing of the pharmacological microglial knockdown.

Strengths

(1) The impact of microglia on hypothalamic synaptic pruning is poorly characterized, and thus, the findings herein are especially of interest.

Weaknesses

(1) Most minor concerns were addressed during revisions, including additional details in the methods and results sections that help interpret the data as presented.

(2) The AgRP staining is unclear. For example, in Figure 2, the figure legend says "labeled AgRP terminals (red)" (Fig 2A-D) but then concludes no difference in the number of "AgRP neurons" (Fig 2J). Is this quantification of AgRP+ neurons, terminals, or both?

(3) The PLX experiments are critical to their conclusion that during lactation, microglia in the PVN sculpt AgRP inputs; however, there is no demonstration that PLX treatment effectively eliminated microglia during this postnatal window. Microglia depletion was only assessed at P55, a month past the PLX treatment window making it unclear when and by what percentage the microglia were eliminated.

---

## [Referee Report · Reviewer #2 (Public review)]

Hypothalamic neural circuits that control body weight develop during the lactation period in rodents. Exposure to maternal high-fat diet during this period (MHFD-L) program has lasting effects on their neuroanatomical organization and function. Microglia sense environmental signals and can sculpt developing circuits by promoting or pruning synaptic connections. Here, the authors examine the contribution of microglia to the effects of MHFD-L to reduce projections from AgRP neurons in the ARH to the PVH, a critical node in circuits regulating energy balance. Using detailed histomorphometric analyses of Iba-1+ cells in the three brain regions (ARH, PVH, and BNST) at two time points (P16 and P30), the authors show that microglial volume and complexity increase, while cell numbers decrease across this period. Exposure to MHFD-L is associated with a transient increase in microglial complexity/volume at P16 in the PVH but not in the other brain regions or time points assessed. Depleting microglia using a pharmacological approach reversed the effects of MHD-L on AgRP outgrowth and body weight.

Strengths:

(1) The Introduction is well-written and provides a good overview of what is known about the roles of microglia in sculpting developing circuits in the hippocampus and cortex. This provides a strong rationale for the current investigations in the hypothalamus.

(2) High-quality imaging and detailed 3-D reconstructions of Iba-1 staining in microglia are used to perform unbiased analyses of microglial complexity and to quantify the spatial relationship between microglial processes and AgRP terminals.

Weaknesses:

(1) The central claim of the manuscript is that microglia in the PVH sculpt the density of AgRP inputs to the PVH in a temporally and spatially restricted manner. While the findings of the microglial ablation experiment are consistent with this hypothesis, they do not prove causality, since their manipulations were not limited to the PVH. Further studies are needed to exclude the possibility that increased outgrowth from AgRP neurons results from direct actions in the ARH or indirect consequences of changes in growth rates.

(2) Impacts of microglial depletion were only assessed in adulthood. Given the hypothesized importance of differences in microglia at P16 and not at P30, it would be helpful to demonstrate that PLX5622 does indeed affect microglia at P16, when the circuit is most sensitive to maternal influences.

---

## [Referee Report · Reviewer #3 (Public review)]

Summary:

The authors interrogated the putative role of microglia in determining AgRP fiber maturation in offspring exposed to a maternal high-fat diet. They found that changes in specific parts of the hypothalamus (but not in others) occur in microglia and that the effect of microglia on AgRP fibers appears to be beyond synaptic pruning, a classical function of these brain-resident macrophages.

Strengths:

The work is very strong in neuroanatomy. The images are clear and nicely convey the anatomical differences. The microglia depletion study adds functional relevance to the paper; however, the pitfalls of the technology regarding functional relevance should be discussed.

Weaknesses:

There was no attempt to functionally interrogate microglia in different parts of the hypothalamus. Morphology alone does not reflect a potential for significant signaling alterations that may occur within and between these and other cell types.

Comments on revised submission: My advice is to change the title by removing "required" and state what is interrogated and found in the paper. A more accurate title would be (for example): Implication of Microglia for Developmental Specification of AgRP Innervation in the Hypothalamus of Offspring Exposed to Maternal High-Fat Diet During Lactation.

I suggest that the authors discuss the limitations of their approach and findings, and propose future directions to address them

---

## [Author Response]

The following is the authors’ response to the original reviews

**Public Reviews:**

**Reviewer #1 (Public reviews):**
(1) A cartoon paradigm of the HFD treatment window would be a helpful addition to Figure 1. Relatedly, the authors might consider qualifying MHFD as 'lactational MHFD.' Readers might miss the fact that the exposure window starts at birth.

This is a good suggestion. The MHFD-L model has been used previously (e.g. Vogt et al. 2014). We have included a cartoon of the MHFD-L model and the PLX treatments to Figure 4, which we feel helps the readers and thank the reviewer for the suggestion.

(2) More details on the modeling pipeline are needed either in Figure 1 or text. Of the ~50 microglia that were counted (based on Figure 1J), were all 50 quantified for the morphological assessments? Were equal numbers used for the control and MHFD groups? Were the 3D models adjusted manually for accuracy? How much background was detected by IMARIS that was discarded? Was the user blind to the treatment group while using the pipeline? Were the microglia clustered or equally spread across the PVN?

In response to this suggestion, we have expanded the description of the image analysis routine in the methods. The analysis focused on detailed changes in microglial morphology as opposed to overall changes in microglia throughout the PVH as a whole. Accordingly, we applied anatomically matched ROIs to the PVH for the measurements. As described in the methods, the Imaris Filaments tool was used to visualize microglia fully contained within a tissue section and a mask derived from the 3D model for these cells was used to isolate them for further analysis, thereby separating these cells from interstitial labeling corresponding to parts of cell processes or other labeling not associated with selected cells. There was no formal “background subtraction.” This was an error in the previous version of the manuscript and we have revised the methods to reflect the process actually used. The images were segmented (to enhance signal to noise for 3D rendering), and then a Gaussian filter was applied to improve edge detection, which facilitates the morphological measurements.

(3) Suggest toning back some of the language. For example: "...consistent with enhanced activity and surveillance of their immediate microenvironment" (Line 195) could be "...perhaps consistent with...". Likewise, "profound" (Lines 194, 377) might be an overstatement.

Revisions have been made to both the Introduction and Discussion to modulate our representation of this controversial issue.

(4) Representative images for AgRP+ cells (quantified in Figure 2J) are missing. Why not a co-label of Iba1+/AgRP+ as per Figure 1, 3? Also, what was quantified in Figure 2J - soma? Total immunoreactivity?

Because the density of AgRP labeling does not change in the ARH we omitted the red channel image (AgRP labeling) to highlight the similarity of the microglial morphology. To address the reviewer’s concerns, in the revised figure we have reconstituted the figure with both the green (microglial) and red (AgRP) channels depicted.

Figure 2J displays the numbers of AgRP neurons counted in the ARH in selected R01s through the ARH. The Methods section has been revised to include the visualization procedure used for the cell counts.

(5) For the PLX experiment:a) "...we depleted microglia during the lactation period" (Line 234). This statement suggests microglia decreased from the first injection at P4 and throughout lactation, which is inaccurate. PLX5622 effects take time, upwards of a week. Thus, if PLX5622 injections started at P4, it could be P11 before the decrease in microglia numbers is stable. Moreover, by the time microglia are entirely knocked down, the pups might be supplementing some chow for milk, making it unclear how much PLX5622 they were receiving from the dam, which could also impact the rate at which microglia repopulation commences in the fetal brain. Quantifying microglia across the P4-P21 treatment window would be helpful, especially at P16, since the PVN AgRP microglia phenotypes were demonstrated and roughly when pups might start eating some chow. b) I am surprised that ~70% of the microglia are present at P21. Does this number reflect that microglia are returning as the pups no longer receive PLX5622 from milk from the dam? Does it reflect the poor elimination of microglia in the first place?

This is an important point and have revised the first sentence in section 2.3 to clarify the PLX treatment logic and added a cartoon to Fig. 4 to show the treatment timeline. The PLX5622 was not administered to the dams but daily to the pups. We also agree with the interpretation that PLX5622 depleted numbers of microglia, as supported by the microglial cell counts, rather than effected a complete elimination and have made revisions to clarify this distinction. Although mice were weighed at weaning, cellular measurements were only made in mice perfused at P55.

(6) Was microglia morphology examined for all microglia across the PVN? It is possible that a focus on PVNmpd microglia would reveal a stronger phenotype? In Figure 4H, J, AgRP+ terminals are counted in PVN subregions - PVNmpd and PVNpml, with PVNmpd showing a decrease of ~300 AgRP+ terminals in MHFD/Veh (rescued in MHFD/PLX5622). In Figure 1K, AgRP+ terminals across what appears to be the entire PVN decrease by ~300, suggesting that PVNmpd is driving this phenotype. If true, then do microglia within the PVNmpd display this morphology phenotype?

We have revised the description of the analysis procedures to clarify these points. All measurements were made in user defined, matched regions of interest according to morphological features of the PVH. No measurements were made that included the entire PVH and we revised the Methods section to improve clarity.

(7) What chow did the pups receive as they started to consume solid food? Is this only a MHFD challenge, or could the pups be consuming HFD chow that fell into the cage?

The pups were weaned onto the same normal chow diet that the dams received prior to MHFD-L treatment. The cages were inspected daily and minimal HFD spillage was observed, although we cannot rule out with certainty any contribution of the pups directly consuming the HFD. We have edited Methods section 5.2 for clarity.

(8) Figure 5: Does internalized AgRP+ co-localize with CD68+ lysosomes? How was 'internalized' determined?

This important point has been clarified by revisions to the Methods section.

(9) Different sample sizes are used across experiments (e.g., Figure 4 NCD n=5, MHFD n=4). Does this impact statistical significance?

Sample size does impact power of ANOVA with larger samples reducing the chance of errors. ANOVA is generally robust in the face of moderate departures from the assumption of equal sample sizes and equal variance such as we experienced in the PLX5622 experiment. Here we used t-tests to detect differences in a single variable between two groups and two-way ANOVA to compare treatment by diet and treatment changes in the PLX5622 studies. Additional detail has been added to the Methods section to clarify this point.

**Reviewer #2 (Public reviews):**
(1) Under chow-fed conditions, there is a decrease in the number of microglia in the PVH and ARH between P16 and P30, accompanied by an increase in complexity/volume. With the exception of PVH microglia at P16, this maturation process is not affected by MHFD. This "transient" increase in microglial complexity could also reflect premature maturation of the circuit.

This is an interesting possibility that requires future investigation (see response to Recommended Suggestions, above).

(2) The key experiment in this paper, the ablation of microglia, was presumably designed to prevent microglial expansion/activation in the PVH of MHFD pups. However, it also likely accelerates and exaggerates the decrease in cell number during normal development regardless of maternal diet. Efforts to interpret these findings are further complicated because microglial and AgRP neuronal phenotypes were not assessed at earlier time points when the circuit is most sensitive to maternal influences.

We agree that evaluations of microglia and hypothalamic circuits at many more time points would indeed be informative (see comments above).

(3) Microglial loss was induced broadly in the forebrain. Enhanced AgRP outgrowth to the PVH could be caused by actions elsewhere, such as direct effects on AgRP neurons in the ARH or secondary effects of changes in growth rates.

A local effect of microglia in the ARH that affects growth of AgRP axons remains a distinct possibility that deserves a targeted examination (see response to Recommended Suggestions, above).

(4) Prior publications from the authors and other groups support the idea that the density of AgRP projections to the PVH is primarily driven by factors regulating outgrowth and not pruning. The failure to observe increased engulfment of AgRP fibers by PVH microglia is therefore not surprising. The possibility that synaptic connectivity is modulated by microglia was not explored.

Synaptic pruning and regulation of axon targeting are not mutually exclusive processes and microglia may participate in both. Here we evaluated innervation of the PVH, which is sensitive to MHFD-L exposure, and engulfment of AgRP terminals by microglia, which does appear to be altered by MHFD-L. Given previous observations of terminal engulfment by microglia in other brain regions in response to environmental changes (e.g. prolonged stress) it is not unreasonable to expect this outcome in the offspring of MHFD-L dams. In future work it will be important to profile multiple cell types in the PVH for microglial dependent and MHFDL-sensitive changes in targeting of AgRP axons. Equally important is a full characterization of postsynaptic changes in PVH neurons.

**Reviewer #3 (Public reviews):**
There was no attempt to interrogate microglia in different parts of the hypothalamus functionally. Morphology alone does not reflect a potential for significant signaling alterations that may occur within and between these and other cell types.The authors should discuss the limitations of their approach and findings and propose future directions to address them.

We agree that evaluations of microglia and hypothalamic circuits at many more time points that include analyses of multiple regions would indeed be informative. We have added statements to the manuscript that address the limitations of our experimental approach and suggest future studies that will extend understanding of underlying mechanisms beyond those investigated here.

**Recommendations for the authors:**

**Reviewing Editors Comments:**
(1) The Abstract is 405 words and should be shortened to less than 200 words.

The abstract has been edited to 200 words.

(2) The authors might consider raising the question in the Introduction of whether reduced AgRP innervation of the PVN in MHFD-treated mice is due to decreased axonal growth, enhanced microglial-mediated pruning, or a combination of both. The potential effects on axonal growth should be given more consideration.

This is an important point that we agree deserves additional consideration in the manuscript. Our past work has focused on leptin’s ability to influence axonal targeting of PVH neurons by AgRP and PPG neurons through a cell-autonomous mechanism and our conclusion is that leptin primarily induces axon growth. Because in this study our design did not focus on changes in axon growth over time but on regional changes in microglia and their interactions with AgRP terminals we did not want to divert attention from our logic in the introduction by highlighting multiple mechanisms. However, we have added a brief mention in the Introduction and have expanded consideration of axonal growth effects to the Discussion. Distinguishing between microglia’s role in synaptic density or axon targeting in this pathway is an important goal of future work.

(3) Line 37, a high-fat diet should be defined here as HFD and used consistently thereafter. Note that "high-fat-diet exposure" requires two hyphens.

The suggested revisions have been made throughout the manuscript.

(4) Line 38 and elsewhere, MHFD does not adequately describe the treatment being limited to the lactation period, perhaps MLHFD would be better or just LHFD (because the pups can't lactate).

The suggested revisions have been made throughout the manuscript, and we have used MHFD-L to describe maternal consumption of a high-fat diet that is restricted to the lactation period.

(5) Line 110, leptin-deficient mice (add hyphen).(6) Line 183, NCD should be defined.

The suggested revisions have been made throughout the manuscript.

(7) Lines 237- 238, it is not clear what is widespread in the rostral forebrain. Is it the loss of microglia? What is the dividing point between the rostral and caudal forebrain? Were microglia depleted in the caudal forebrain too?

We have revised this section of the manuscript to focus the description on the hypothalamus alone and specify that the reduction in microglial density is not restricted to the PVH.

(8) Line 245, microglial-mediated effects (add hyphen).(9) Line 247, vehicle-treated mice (add hyphen).

The suggested revisions have been made throughout the manuscript.

(10) Line 457, when referring to genes, the approved gene name should be used in italics, AgRP should be Agrp (italics).

The suggested revision has been made throughout the manuscript.

(11) Line 459, the name of the Syn-Tom mice in the Key Resource table, Methods, and Text should be consistent. It would be best to use the formal name of the Ai34 line of mice on the JAX website.

The suggested revisions have been made throughout the manuscript.

(12) Figure 1G H, and I um should have Greek micro; Fig. 1J and K, Replace # with Number. The same suggestions apply to all the other figures.

Both the manuscript and figures have been revised in accordance with this recommendation.

(13) Figures 4 G, H, I and J. and Figures 5 M and O. The font size is too small to see well.

Fonts have been changed in the figures to improve visibility.

**Reviewer #1 (Recommendations for the authors):**
(1) Figures are out of order in the text. For example, Figure 1A is followed next by the results for Figure 1J instead of Figure 1B.

We regret that the organization of figure panels makes for awkward matching for the reader as they proceed through the text. We designed the figures to facilitate comparisons between cellular responses and differences in labeling. After evaluating a reorganization, we decided to maintain the original panel configurations, but have revised the text to more closely follow the presentation of cellular features in the figures.

(2) Figure 1B.: All images lack scale bars.(3) Line 433 - 'underlie' is spelled wrong.(4) Rosin et al should be 2019 and not 2018.

These corrections have been implemented in the revised text and figures.

(5) The statement that "the effects of MHFD on microglial morphology in the PVH of offspring display both temporal and regional specificity, which correspond to a decrease in the density of AgRP inputs to the PVH" (Line 196) needs clarification, as the phrase "regional specificity" has not been substantiated in this section even though it is discussed later.

We agree with this comment and have revised section 2.1 to more closely match the data presented to this point in the manuscript.

**Reviewer #2 (Recommendations for the authors):**
(1) The claim of "spatial specificity" in the effects of MHFD on microglia is based on an increase in the complexity/volume of microglia at P16 in the PVH that was not seen in the ARH or BNST. The transient nature of the effect raises several questions: Does the effect on the PVH represent premature maturation?

This is an interesting suggestion. However, given how little is known about microglial maturation in the hypothalamus it is difficult to address. It is indeed possible that microglia mature at different rates in each AgRP target, and that MHFD-L exposure alters the rate of maturation in some regions but not others. This will require a great deal more analysis of both microglia and ARH projections to understand fully (see below).

(2) To support their central claim that microglia in the PVH "sculpt the density of AgRP inputs to the PVH" the authors report effects on Iba1+ cells in the PVH of chow-fed dams at P55, body weight at P21, and AgRP projections in the PVH at an unspecified age. It is hard to understand what is happening across "normal" development in chow-fed dams since the number of Iba1+ cells decreases from ~50 to ~25 between P16 and P30 (Figure 1), but then increases to >60 cells at P55 (Figure 4). Given the large fluctuations in microglial population across time, analyzing the same parameters (i.e. microglial number/morphology in the ARH and PVH, AgRP neuronal number in the ARH, and fiber density in the PVH, and body weight) across time points before, during and after the critical period in chow and MHFD conditions would be very helpful.

The time points we evaluated were chosen to be during and after the previously determined critical period for development of AgRP projections to the PVH, which were then compared with adults (which were all P55) to assess longevity of the effects. We have incorporated revisions to improve the clarity of when measurements were assessed, and treatments implemented. Defining the cellular dynamics of microglia across time remains a major challenge for the field and will certainly be informed by future studies with additional time points, as well as by in vivo imaging studies focused on regions identified here. Although such studies are beyond the scope of the present work, their completion would advance our current understanding of how microglia respond to nutritional changes during development of feeding circuits.

(3) As microglia are also ablated in the ARH, direct effects on AgRP neurons or indirect effects via changes in growth rates could also contribute to increased AgRP fiber density in the PVH. In support of the first possibility, postnatal microglial depletion increases the number of AgRP neurons (Sun, et al. 2023).

We agree with the suggestion, also raised by the Reviewing Editor, which has been addressed briefly in the Introduction, and in more detail by revisions to the Discussion section.

(4) The failure to assess alpha-MSH fibers in the same animals was a missed opportunity. They are also affected by MHFD but likely involve a distinct mechanism (Vogt, et al 2014).

Given the paired interest in POMC neurons and AgRP neurons I understand the reviewer’s comment. We chose to focus solely on AgRP neurons because we do not currently have a way to genetically target axonal labeling exclusively to POMC neurons due to the shared precursor origin of POMC neurons and a percentage of NPY neurons in the ARH, as shown by Lori Zeltser’s laboratory. Moreover, the elegant work by Vogt et al. focused on responses of POMC neurons in the MHFD-L model. However, it certainly remains possible that microglia in the PVH interact with terminals derived from POMC neurons, as well as with terminals derived from other afferent populations of neurons.

(5) All statistical analyses involved unpaired t-tests. Two-way ANOVAs should be used to assess the effects of age and HFD and interactions between these factors.

We used t-tests to detect differences in a single variable between two groups and two-way ANOVA to compare treatment by diet and treatment changes in the PLX5622 studies. Additional detail has been added to the Methods section and information added to the figure legend for Fig. 4 to clarify this point.

**Reviewer #3 (Recommendations for the authors):**
I suggest exploring the deeper characterization of the microglia in various parts of the hypothalamus in different conditions. This could include cytokine assessment or spatial transcriptomic.

We agree that a great deal more work is needed to improve our understanding of how microglia impact hypothalamic development more broadly and to identify underlying molecular mechanisms. We are hopeful that the data presented here will motivate additional study of microglial dynamics in multiple hypothalamic regions, as well as detailed studies of cellular signaling events for factors derived from MHFD-L dams that impact neural development in the hypothalamus.